



# Measurements from mobile surface vehicles during LAPSE-RATE

Gijs de Boer[1,2], Sean Waugh[3], Alexander Erwin[4], Steve Borenstein[5], Cory Dixon[5*], Wafa'a
5 Shanti[4], Adam Houston[4], Brian Argrow[5]

[1] Cooperative Institute for Research in Environmental Sciences, University of Colorado Boulder, Boulder, Colorado, USA
[2] NOAA Physical Sciences Laboratory, Boulder Colorado, USA
10 [3] NOAA National Severe Storms Laboratory, Norman, Oklahoma, USA
[4] University of Nebraska – Lincoln, Lincoln, Nebraska, USA
[5] Integrated Remote and In Situ Sensing, University of Colorado Boulder, Boulder, Colorado, USA
* Now at Geotech Environmental Equipment, Denver, Colorado, USA

*Correspondence to*: Gijs de Boer (gijs.deboer@colorado.edu)

**Abstract.** Between 14 and 20 July 2018, small unmanned aircraft systems (sUAS) were deployed to the San Luis Valley of Colorado (USA) alongside surface-based remote, in-situ sensors, and radiosonde systems as part of the Lower Atmospheric Profiling Studies at Elevation – a Remotely-piloted Aircraft Team Experiment (LAPSE-RATE). The measurements collected as part of LAPSE-RATE targeted quantities related to enhancing our understanding of 20 boundary layer structure, cloud and aerosol properties and surface-atmosphere exchange, and provide detailed information to support model evaluation and improvement work. Additionally, intensive intercomparison between the different unmanned aircraft platforms was completed. The current manuscript describes the observations obtained using three different types of surface-based mobile observing vehicles. These included the University of Colorado Mobile UAS Research Collaboratory (MURC), the National Oceanic and Atmospheric Administration National 25 Severe Storms Laboratory Mobile Mesonet, and two University of Nebraska Combined Mesonet and Tracker (CoMeT) vehicles. Over the one-week campaign, a total of 143 hours of data were collected using this combination of vehicles. The data from these coordinated activities provide detailed perspectives on the spatial variability of atmospheric state parameters (air temperature, humidity, pressure, and wind) throughout the northern half of the San Luis Valley. These data sets have been checked for quality and published to the Zenodo data archive under a specific 30 "community" set up for LAPSE-RATE (https://zenodo.org/communities/lapse-rate/) and are accessible at no cost by all registered users. The primary dataset DOIs are 10.5281/zenodo.3814765 (CU MURC measurements; de Boer et al., 2020d), 10.5281/zenodo.3738175 (NSSL MM measurements; Waugh, 2020) and 10.5281/zenodo.3838724 (UNL CoMeT measurements; Houston and Erwin., 2020).



## 1 Background

In July 2018, a collection of atmospheric scientists and engineers from around the globe converged on the San Luis Valley (SLV) of Colorado (USA) to take part in the LAPSE-RATE (Lower Atmospheric Profiling Studies at Elevation – a Remotely-piloted Aircraft Team Experiment) field campaign (de Boer et al., 2020a; 2020b). This campaign was focused on demonstrating the utility of unmanned aircraft systems (UAS) for atmospheric research and collecting scientifically-interesting data sets to conduct targeted studies on specific topics of interest related to boundary layer

processes. Connected to the annual meeting of the International Society for Atmospheric Research using Remotely-piloted Aircraft (ISARRA, de Boer et al., 2019), LAPSE-RATE included over 100 scientists and engineers, who together conducted nearly 1300 research flights and captured over 250 flight hours of data using UAS.

Information on the different UAS and profilers deployed during LAPSE-RATE, numerical simulations completed for

the campaign, and an overview of the campaign itself is distributed in a range of articles, many of which are associated with this *Earth System Science Data* special issue and will not be revisited here (de Boer et al., 2020b; de Boer et al., 2020c; Bell et al., 2020; Pillar-Little et al., 2020; Bailey et al., 2020; Natalie et al., 2020; Islam et al., 2020; Brus et al., 2020; Pinto et al., 2020). The current paper is focused on describing datasets collected using mobile surface observing systems during the LAPSE-RATE campaign.


The general concept behind mobile surface observing vehicles is to provide a fully-mobile platform from which accurate observations of atmospheric parameters can be made, thereby offering opportunities to position (and reposition) in situ surface meteorological instrumentation precisely to capture highly-localized gradients and target locations that are thought to be critical for development of development of phenomena of interest. To accomplish

this, rack- or mast-mounted instrumentation is set up to measure quantities such as pressure, wind speed/direction, temperature, and relative humidity. In the case of vehicles set up for truly mobile measurements (i.e. measuring while driving) instruments are mounted far enough away from the vehicle structure to minimize direct influence to the observations from the vehicle itself. Such mobile mesonet systems have been deployed for atmospheric research for over two decades and details on the original mobile mesonets can be found in Straka et al. (1996). These systems

have generally been used to evaluate atmospheric conditions supporting the development of tornadic supercells (e.g. Markowski, 1999; Pietrycha and Rasmussen, 2004), though deployments to observe land-falling hurricanes have also been conducted (e.g. Caban et al., 2019). Additionally, mobile mesonet systems have been used in conjunction with airborne systems (manned or unmanned) to capture measurements along four-dimensional transects (e.g. Riganti and Houston 2017).


During LAPSE-RATE, atmospheric-observing surface vehicles provided critical insight into gradients in state variables across the SLV, including information on temperature, pressure, humidity and winds. This includes both single-site sampling, as well as measurements covering extended transects conducted throughout the northern half of the SLV. The latter were particularly interesting given that the SLV features widely-varying surface types (ranging

from irrigated cropland to dry shrublands), significant topography, and terrain-induced flows. The following section



describes the vehicles deployed and instrumentation that they carried, while section 3 provides an overview of measurement locations and sampling strategies. Section 4 provides details on data processing and quality control, while section 5 offers information on dataset availability.

## 2 Instrument and Vehicle Descriptions

The three mobile surface-based platforms used to collect data during LAPSE-RATE and described in this paper include the Mobile UAS Research Collaboratory (MURC), operated by the University of Colorado Boulder (CU), a mobile mesonet (MM) vehicle operated by the NOAA National Severe Storms Laboratory (NSSL), and a pair of Combined Mesonet and Tracker (CoMeT) vehicles operated by the University of Nebraska Lincoln (UNL). As documented below, these three vehicles provided complementary measurements, including details on atmospheric

temperature, pressure, humidity and winds. Each operated in slightly different modes over the duration of the campaign, though all three vehicles were at the same location for limited times.

### 2.1 CU MURC

The CU MURC (figure 1) is an instrumented van that was added to the Integrated Remote and In-Situ Sensing (IRISS) program vehicle fleet in early 2018. This system was specifically developed to work in tandem with unmanned aircraft

operations, serving as a mobile command station and surface measurement facility during field deployments. This centralized operations center provides a platform from which to oversee field teams and provide general situational awareness. The MURC is equipped with two workstations for lighter computing loads, including on-site processing of data, real-time communications with team members and the broader community through web-based systems, and possibly serving as UAS ground stations. Additionally, the MURC carries two servers for more intensive computing

tasks, with one dedicated to graphics intensive processes (such as processing of imagery for photogrammetry-centric missions) and the other dedicated to general computing and intensive real-time data processing.

From an observational perspective, the MURC is equipped with a 15 m extendable mast, atop which are mounted several meteorological sensors. This includes a Gill MetPak Pro Base Station that measures barometric pressure, air

temperature, and relative humidity, a Gill WindMaster 3D sonic anemometer for 3D wind and fast temperature measurements, and an R.M. Young Wind Monitor (05103) propeller and vane anemometer which provides a redundant horizontal wind measurement and offers real-time situational awareness for nearby unmanned aircraft operators. An overview of the sensors and their projected accuracies is included in Table 1. The MURC is also equipped with a large communications suite that increases the range of UHF/VHF vehicle-to-vehicle radios used

during field campaigns, increases cellular bandwidth for data transfer and communications, and improves the ground station to UAS communication link. While mobile, the MURC is set up to operate at a single location at any given time, and does not collect measurements while travelling like other platforms described below. Data collected by the MURC were used to intercompare measurements from the different UAS deployed during LAPSE-RATE. The results of this intercomparison are documented in Barbieri et al., (2019).




## 2.2 NSSL Mobile Mesonet

In addition to the MURC, LAPSE-RATE included a deployment of the NOAA NSSL MM vehicle, a heavily-modified version of the original MM created 25 years ago (Straka et al., 1996). The current generation NSSL MM is built on a Ford F-250 extended cab long bed pickup truck. Instrumentation is located on a rack mounted forward and above the hood of the vehicle, in order to minimize atmospheric disturbances caused by the blunt forward edge of the truck (figure 2). Mounting the equipment rack over the roof of the vehicle (as was done with previous NSSL MM vehicles) was thought to result in observational biases due to the turbulent and accelerated airflow over the vehicle roof and thermal influence of the vehicle engine. The new set up requires the addition of a substantial structure to support the weight and drag of the instrument rack. This structure also allows for installation of a wire mesh hail cage to protect the windshield from hail strikes while operating in the vicinity of severe thunderstorms. In addition to the instrument rack, the MM can carry up to four helium tanks along with a Vaisala MW41 sounding system for mobile radiosonde launches.

For air temperature and relative humidity measurements, the NSSL MM deploys a Vaisala HMP155 sensor. While the HMP155 is highly accurate, the relative humidity sensor can be prone to contamination by atmospheric particles. To reduce this contamination the HMP155 requires integration of a membrane that allows water molecules to pass through while reducing the impact of contaminants. While this practice protects the RH observations from contamination, it also significantly delays the thermal response of the environment inside the membrane. In combination with a relatively slow time constant (testing on previous models such as the HMP35 and HMP45 have revealed time constants on the order of 10 mins (Waugh, 2012)), the impacts of this membrane result in very slow response temperature measurements (hence the designation "slow temp" for the HMP155). To overcome this, a fast responding Campbell Scientific T109SS temperature sensor is also installed on the NSSL MM (T109, aka "fast temp"). The HMP155 cannot be assumed to report a temperature and relative humidity that represents the true environmental conditions, only the conditions inside the membrane. However, while the HMP should not be used for temperature and relative humidity observations directly, dewpoint is conserved across the membrane allowing the HMP to be useful for observing the dewpoint (Richardson et al. 1998). This dewpoint observation is combined with data from the faster T109 temperature sensor to derive a relative humidity value that is representative of the true environmental value (Richardson et al., 1998). The thermodynamic observations are housed in a radiation shield to protect the sensors from direct solar radiation while maintaining adequate airflow from the real environment. This shield, known as the "U-tube", was developed by NSSL to specifically accomplish this task (Waugh and Frederickson 2010; Houston et al. 2016). For wind measurements, the NSSL MM deploys a standard propeller-vane combination anemometer from RM Young (Wind Monitor 05103) covering a wide range of wind speeds (0-100 ms$^{-1}$). While the vehicle is stationary, ambient wind direction is derived using the vehicle-relative wind direction and vehicle heading from a KVH C100 magnetic compass. In combination, these allow for computation of the ambient wind vector. While in motion, the vehicle-relative wind vector is subtracted from GPS-obtained vehicle motion to produce the inertial environmental wind vector. The measurements from all of these sensors are logged at 1Hz using a Campbell Scientific



CR6 wifi-enabled data logger. A list of the sensors and their respective measurements, as well as general accuracies and response times are listed in Table 2.

### 2.3 UNL CoMeT

Finally, the UNL deployed two CoMeTs for LAPSE-RATE. These systems are built around Ford Explorers and feature a forward-mounted suite of meteorological sensors and a dual moonroof (see Figure 3). The CoMeTs measure air temperature and relative humidity at ~2 m above ground level (AGL) using a Vaisala HMP155A and also include a fast-response sensor (Campbell Scientific 109SS thermistor) for air temperature at ~2 m AGL (same set up as the NOAA NSSL MM). Air pressure at ~2.5 m AGL is measured using a Vaisala PTB210, while wind speed and direction

are observed using an R.M. Young 05103 propeller-vane anemometer at approximately 3.5 m AGL. The vehicle heading is tracked using a KVH Industries C-100 fluxgate compass. As on the NSSL MM, the HMP155A and 109SS thermistor are shielded and aspirated within a U-tube (Waugh and Frederickson 2010; Houston et al. 2016). Manufacturer specifications for these instruments are provided in Hanft and Houston (2018) and are again listed in Table 3 of this manuscript. In addition to the measured variables, the CoMeT data loggers (Campbell Scientific CR6)

along with a custom Python script, use observed quantities to calculate dewpoint temperature ($T_d$), mixing ratio ($q_v$), potential temperature ($\theta$), equivalent potential temperature ($\theta_e$), virtual potential temperature ($\theta_v$), and wind speed and direction. The equations used to compute these quantities are provided in section 4.

### 3 Description of measurement locations, deployment strategies and sampling

The vehicles described above covered a significant amount of ground over the course of the campaign. Each played

a different role in addressing the primary objectives of the LAPSE-RATE campaign (see de Boer et al., 2020a; 2020b). These vehicles were used to evaluate the performance of UAS sensors, in addition to intercomparison between different surface vehicles (see Figure 4). Figure 5 provides an overview of the amount of time each vehicle spent making atmospheric measurements during the campaign. As shown, sampling primarily occurred in the morning and early afternoon (local time), with the NSSL MM generally starting the earliest in order to launch an early morning

radiosonde (see Bell et al., 2020). Sampling conducted on the afternoon of 14 July and the morning of 20 July by the CU MURC was in support of platform intercomparison efforts (Barbieri et al., 2019).

The primary role of the CU MURC was to provide daily measurements at a consistent location (Leach Airport) in the center of the sampling domain. In this role, the MURC acted as a meteorological tower that collected measurements

in a similar manner from day to day, providing a baseline for putting other observations collected during the campaign into context. The only exception to this routine sampling took place on July 19, when all platforms were focused on cold air drainage out of the smaller valleys on the northern end of the SLV. On July 19, the MURC was positioned a bit farther to the north, as can be seen in Figure 6f to help evaluate the timing and intensity of density currents flowing from Saguache (northwest corner of the SLV) and Villa Grove (northeast corner of the SLV). In total, the MURC

operated for seven days, capturing a total of 45.5 hours of data.



The NSSL MM filled multiple roles throughout the campaign. One important role included the launching of radiosondes from various locations around the SLV (Bell et al., 2020). This often included early morning radiosondes from Leach Airport. In addition, the NSSL MM was leveraged as a mobile measurement platform to capture
information on spatial variability throughout the broader valley. The first of these mobile measurement sorties took place on 15 July and included transects spanning the area between Alamosa, Colorado and Moffatt, Colorado. These transects covered a variety of different surface types, ranging from irrigated cropland to dry desert-like areas on the eastern side of the SLV. On 16-18 July, the NSSL MM focused on the south central portion of the SLV, with much of the measurement time spent at Leach Airport, and the transect between Leach Airport and the city of Alamosa.
Finally, on 19 July, the NSSL MM covered area from Alamosa to Saguache in the northwest corner of the SLV. Most of the time on that date was spent sampling the square shown in the northwest part of the SLV in Figure 6f to help understand the spatial variability of the drainage flow exiting the Saguache Valley. In total, the NSSL MM collected a total of 55.4 hours of data, in addition to the measurements from the radiosondes launched.

The two UNL CoMeT vehicles were deployed separately throughout the San Luis Valley during the majority of the LAPSE-RATE campaign. CoMeT-1 was principally focused on coordinated observations with the CU UAS team and involved both stationary data collection based at the Leach Airport and transect data collection across the SLV. CoMeT-2 was principally focused on stationary data collection in coordination with the UNL UAS team based at a site on the eastern margins of the SLV in the northwest corner of the Great Sand Dunes National Park (henceforth
referred to as observation site "Gamma"; Islam et al., 2019; 2020). As with the NSSL MM, the CoMeT data collection began on 15 July with CoMeT-1 operating transects based out of Leach Airport and CoMeT-2 collecting stationary observations at Gamma. Similar operations were executed on 16 and 18 July during which time CoMeT-1 performed extended east-west transects to the far eastern portion of the SLV to help understand the role of surface type gradients and sloping terrain on that side of the valley on convection initiation. On 18 July, following operations at Gamma,
CoMeT-2 also executed a set of transects along the eastern margins of the irrigated region of the valley in an effort to explore whether surface flow parallel to this margin resulted in a coherent convergence boundary. On 17 July, both CoMeTs operated at the Leach Airport. Finally, on 19 July, both CoMeTs joined the effort to capture the early morning Saguache Valley cold-air drainage, with frequent transects along County Road X between Saguache and County Road 55. Over the course of the campaign, CoMeT-1 collected 50.4 hours of data and CoMeT-2 collected
50.3 hours of data.

Figure 7 provides a statistical overview of data collected by these three platforms over the duration of the LAPSE-RATE campaign. Included are normalized probability distributions of measured quantities, including temperature, relative humidity, air pressure, wind speed, wind direction and, for the NSSL MM and UNL CoMeT datasets, the
difference between the fast and slow temperature sensors. For all of these distributions, data were averaged to a moving 1-minute equal weighted window. The distributions illustrate differences that are likely largely the result of instrument and platform location. For example, it is important to remember that while the NSSL MM and UNL CoMeT instruments were located close to (<3 m) the ground, the CU MURC data were collected atop a 50-foot mast.



Therefore, it is not surprising that the CU MURC pressure measurements are found to be slightly lower than those
measured by the other two platforms. Similarly, the CU MURC temperature and RH distributions lack the extremes,
with measurements from the top of the mast likely missing the coldest temperatures in the early morning, and the
warmest temperatures in the afternoon. Also attributable to this altitude difference is the slight but noticeable increase
in wind speeds and counterclockwise shift in wind direction from the near surface environment to the height of the
mast. Finally, for both the NSSL MM and the UNL CoMeT data, the slow temperature sensor inside of the membrane
(HMP155) was shown to have a warm bias relative to the fast sensor (T109 SS), and the UNL CoMeT difference
distribution is shown to have an extended tail towards positive values. The mean difference (fast minus slow) of the
NSSL MM temperature sensors was -0.328 C, while the mean difference of the UNL CoMeT temperature sensors was
-0.398 C.

Figure 8 provides additional insight into the temporal variability of the recorded variables, both in terms of diurnal
cycle and over the extent of the LAPSE-RATE campaign, based on measurements obtained by the CU MURC. The
upper left-hand panel shows that, as expected, temperatures were generally coldest in the early morning, with a gradual
but notable warming over the course of the day. Along with this, relative humidity levels were generally highest in
the morning and decreasing significantly over the course of the day. Interestingly, the middle of the week did feature
one day (17 July) where the MURC was sampling later into the afternoon, and temperatures were recorded dropping
during that time period, decreasing from around 26 C in the mid-afternoon to below 20 C by the end of sampling
around 1700 MDT. The upper right-hand panel provides insight into the variability occurring over the course of the
field campaign. The earliest days (07/14-07/16) were consistently warm and relatively humid. The atmosphere
became drier later in the campaign, with relative humidity values peaking at around 70% on 19 July, despite
temperatures that were slightly cooler than from those recorded on 17 and 18 July, when relative humidity levels
climbed above 90% in the early morning hours. In general, afternoons were illustrated to fall between 25-27 C, and
mornings between 10-15 C (largely depending on the start time of sampling for a given date). Wind speed and
direction measurements are shown in the bottom panel of Figure 8 and demonstrate that winds were generally quite
light throughout the week, with values between 0-6 m s$^{-1}$. July 14, 15 and 17 did see an intensification of winds in
the afternoon, generally associated with convective systems developing over the valley and surrounding mountain
peaks. Wind directions were generally from the south and east. With the stronger winds resulting in a more northerly
and westerly component. The only sampling period with solidly westerly winds was the morning of 20 July.

## 4 Data processing and quality control

### 4.1 CU MURC

Data available from the CU MURC have been processed in various ways to average the data, remove outliers, and
correct the wind measurements from the sonic anemometer for platform pitch and roll. All data were averaged across
a moving 1-second window. Any data points falling inside of the 1-second window were included in averaging,
though no filter was implemented to ensure any particular number of samples within a given 1-second averaging
window. Time periods where no data were collected are included as "NaN". Screening for outliers was completed



using the Matlab *filloutliers* function, which detects and replaces outliers using a linear interpolation between points not deemed to be outliers. In the current application, outliers were defined as points falling more than three local standard deviations outside of a moving mean window encompassing 10 seconds worth of data. Note that this technique was applied to the measured zonal and meridional wind components only, and not to the wind speed and direction included in the dataset that are calculated using the components, given that such averages are not possible

on the vector values.

Rotation of the CU MURC sonic anemometer data was completed using a standard three axis rotation (Tropea et al., 2007), where the updated wind coordinates are calculated as follows:

$$\begin{bmatrix} u_f \\ v_f \\ w_f \end{bmatrix} = \mathbf{A} \begin{bmatrix} u_m \\ v_m \\ w_m \end{bmatrix}$$

Here, $u_m$, $v_m$ and $w_m$ are the measured instantaneous velocity components as measured by the sonic anemometer, $\mathbf{A}$ is the rotation matrix, and $u_f$, $v_f$ and $w_f$ are the final velocity components. $\mathbf{A}$ can be approximated by combining multiple rotations to align the coordinate system using measured Euler angles. In this case, we assume:

$$\mathbf{A} = \mathbf{T} \cdot \mathbf{S} \cdot \mathbf{R}$$

where:

$$\mathbf{T} = \begin{bmatrix} 1 & 0 & 0 \\ 0 & \cos\psi & \sin\psi \\ 0 & -\sin\psi & \cos\psi \end{bmatrix} \mathbf{S} = \begin{bmatrix} \cos\varphi & 0 & \sin\varphi \\ 0 & 1 & 0 \\ -\sin\varphi & 0 & \cos\varphi \end{bmatrix} \mathbf{R} = \begin{bmatrix} \cos\theta & \sin\theta & 0 \\ -\sin\theta & \cos\theta & 0 \\ 0 & 0 & 1 \end{bmatrix}$$


and $\psi$, $\varphi$, and $\theta$ are the roll, pitch and yaw rotation angles, respectively, as measured by the CU MURC operators. Note that these angles were only measured once after parking the vehicle and do not vary in time in between vehicle movements. Therefore, any swaying of the vehicle as a result of people getting in and out, wind, or for other reasons may impact the wind measurements from the sonic anemometer and may not be accounted for.

Note that the rotations are applied in yaw, pitch, roll order, meaning that we step through the rotation as follows:

$$\begin{aligned} u_1 &= u_m \cos\theta + v_m \sin\theta \\ v_1 &= -u_m \sin\theta + v_m \cos\theta \\ w_1 &= w_m \end{aligned}$$

$$\begin{aligned} u_2 &= u_1 \cos\varphi + w_1 \sin\varphi \\ v_2 &= v_1 \\ w_2 &= -u_1 \sin\varphi + w_1 \cos\varphi \end{aligned}$$


$$\begin{aligned} u_f &= u_2 \\ v_f &= v_2 \cos\psi + w_2 \sin\psi \\ w_f &= -v_2 \sin\psi + w_2 \cos\psi \end{aligned}$$




Note that these corrections are only applied to the wind measurements from the sonic anemometer. The influence of slight offsets in pitch and roll are negligible for the R.M. Young propeller-based wind instrument as the propeller follows a cosine response. For LAPSE-RATE, the sensor pitch and roll angles varied between -1.7 and 2.35 degrees and -2.21 and 3.48 degrees, respectively. These angles correspond to a maximum error of 0.2%, well below the uncertainty of the instrument. The sensor was aligned with magnetic north on a daily basis.

**4.2 NSSL MM**

For the NSSL MM's, a majority of the data processing and variable calculation is done in real time on the CR6 data logger. Most of the observations do not require much in the way of modification, the exception to that are derived ambient winds, vehicle heading, and the environmental RH. This is an advantage of the CR6 datalogging system as the onboard computing power is enough to handle the calculations in real time, making data display and recording easier.

For the derived winds, the measured wind speed and direction directly off the anemometer is a combination of the vehicle motion vector and the ambient wind vector (this combined vector is the vehicle relative vector), which need to be separated. To obtain the ambient wind vector, the vehicle motion must be subtracted from the vehicle relative vector in a process similar to that outlined in Section 4.1 for the CU MURC, though not as complex as the wind monitor on the NSSL MM is only two dimensional. The vehicle motion obtained from the onboard GPS and broken into N-S and E-W components. The apparent wind vectors as measured by the anemometer directly is also broken into components, but is first added to the vehicle heading to obtain a true directional vector rather than a vehicle relative vector. The apparent wind components are then subtracted from the vehicle motion components to obtain the ambient wind components. The final step of the process involves converting the ambient components back into vector form, which requires a tedious series of manual computations to determine the quadrant relative angle and its true meteorological heading. The vector wind speed is found simply with:

$$wind\_speed \ = \ \sqrt{X^2 + Y^{\wedge}2} \ ,$$

where X and Y are the ambient wind components for the U and V directions respectively. For the vector wind direction, the components must be examined to determine where on the meteorological coordinate system they lie and manually assembled in the correct direction. This is due to the fact that traditional use of sin, cos, and tangent (and their inverse functions) are referenced to a mathematical coordinate system which is reversed and 90° offset from the meteorological coordinate system. To determine the wind direction, an offset to either 90° or 270° is found by taking the ATAN(ABS(Y/X)). This value is then added or subtracted from the appropriate reference angle depending on the quadrant. For example, if Y and X were both +15 ms[-1], then ATAN(ABS(Y/X)) = 45°. Since both Y and X are positive values, the resulting angle should be in the first quadrant, or between 0° and 90°, thus the ATAN value is subtracted from the 90° reference angle, obtaining an environmental wind direction of 45°. A more detailed description of this process is forthcoming in a future manuscript. Note that if the vehicle is not moving, this component based approach is not needed and the wind direction can be found by simply rotating the observed winds by the vehicle heading while stationary.

The vehicle heading is also corrected in real time for cases where the vehicle is not moving. In these situations, the heading of the vehicle is obtained via a magnetic compass which provides the magnetic bearing. This is used in cases where the vehicle motion is less than 1 ms[-1]. The magnetic heading differs from true north by an offset which is dependent on the coordinates of the observation location, called the magnetic declination angle. This angle is provided along with the GPS coordinates in real time, and is used to correct the magnetic heading.


While the NSSL MM measures temperature and humidity, it does so with a set of sensors behind a protective membrane that significantly delays the response time as described in Section 2B. With this filter in place, the measured RH is lagged behind the true environmental RH and must be rederived. This process follows that of Richardson et al. (1998) where:


$$Derived_{RH} = \frac{e}{e_s} * 100$$

$$e = 6.1365 * EXP(\frac{17.502 * TdC}{240.97 + TdC})$$

$$e\_s = 6.1365 * EXP(\frac{17.502 * Tfast}{240.97 + Tfast})$$

where Tfast and TdC (the calculated dewpoint from the HMP155) are in Celsius. The calculation for dewpoint is done

with a built in CR6 function for dewpoint, which uses Tetens' equation and the vapor pressure (Campbell Scientific, 2020).

As a final step to the process, after the data is collected and archived, each data set is run through a QC procedure where the individual data files from a single operations period are combined, and a set of QC flags applied. The intent

of these flags are not to remove data, but rather flag data that is potentially suspicious and should be examined manually. There are four QC flags, representing: panel temperature excess, vehicle stationary periods, excessive changes to vehicle motion, and a general sanity check. More specific details of the QC flags are contained in the readme files that accompany the data, however a brief description is presented here. The panel temperature flag identifies areas where the internal temperature of the CR6 datalogger changes by a significant amount. This identifies

periods where the logger may be having inconsistency issues or power supply problems, which manifest themselves in the internal temperature monitoring first. The second QC flag is meant to identify periods where the vehicle is stationary, which could increase the potential for bias in the observations, while the third flag looks for sharp changes to the speed or direction. The latter flag is meant to identify areas where there could be a discrepancy between the vehicle heading and the observed winds, such as in a sharp turn. The final QC flag simply examines all the observations

for values that are well outside the normal operating range.

**4.3 UNL CoMeTs**





As mentioned in Section 2, the CoMeT data loggers and Python scripts are used to calculate key quantities of interest
in real-time. These quantities include corrected/fast relative humidity, water vapor mixing ratio, dew point
temperature, the potential temperature ($\theta$), virtual potential temperature ($\theta_v$), and equivalent potential temperature
($\theta_e$),.

For both CoMeT-1 and CoMeT-2, relative humidity is adjusted to the fast temperature following Richardson et al.
(1998) and Houston et al. (2016): vapor pressure is calculated using slow temperature and relative humidity, saturation
vapor pressure is calculated using fast temperature, and the ratio of the two is used to calculate the corrected/fast
relative humidity. In CoMeT-1, the calculations are done in the Python script using the following:

$$e_* = 6.112 \exp\left[\frac{17.67 \cdot T_*}{243.5 + T_*}\right],$$

from Wexler (1976) and Bolton (1980) where $e_*$ is either vapor pressure or saturation vapo pressure and $T_*$ is dew
point temperature (for vapor pressure) or fast temperature (for saturation vapor pressure). Dew point temperature is
calculated using

$$T_d = \frac{257.14\gamma}{18.678 - \gamma}$$

$$\gamma = \ln\left(0.01 \cdot RH_*\right) + T_{slow}\frac{18.678 - T_{slow}/234.5}{257.14 + T_{slow}}$$

Where $RH_*$ is the uncorrected (slow) relative humidity and $T_{slow}$ is the slow temperature. In contrast to CoMeT-1,
the calcluation of dew point temperature, vapor pressure, and saturation vapor pressure are done within the logger,
and slightly different expressions are used. For dew point temperature:

$$T_d = \frac{-5420}{\ln\left(\frac{p \cdot q_v}{62.2 \cdot 2.53 \times 10^9}\right)}$$

$$T_d = \frac{A_3 \ln\left(e/A_2\right)}{A_2 - \ln\left(e/A_1\right)}$$

is used, where $A_1 = 0.61078$, $A_2 = 17.558$, and $A_3 = 241.88$. The expression used in the logger for (saturation)
vapor pressure is from Lowe (1977):

$$e_* = B_0 + B_1T + B_2T^2 + B_3T^3 + B_4T^4 + B_5T^5 + B_6T^6$$

where, $B_0 = 6.107799961$, $B_1 = 4.436518521 \times 10^{-1}$, $B_2 = 1.428945805 \times 10^{-2}$, $B_3 = 2.650648471 \times 10^{-4}$,
$B_4 = 3.031240396 \times 10^{-6}$, $B_5 = 2.034080948 \times 10^{-8}$, and $B_6 = 6.136820929 \times 10^{-11}$.

Water vapor mixing ratio is calculated using

$$q_v = 62.2\frac{e}{p}$$



and dew point temperature and vapor presure as described above. Potential temperature for both CoMeTs is calculated using

$$\theta = T_{fast}\left(\frac{10^5}{p}\right)^{R_d/C_{pd}}.$$

Virtual potential temperature is calculated using

$$\theta_v = \theta\left(1 + 0.61q_v\right).$$

Equivalent potential temperature is calculated in both CoMeTs following Bolton (1980):

$$\theta_e = T_m \exp\left[\left(\frac{3376}{T_{LCL}} - 2.54\right)q_v\left(1 + 0.81q_v\right)\right]$$

$$T_m = \theta\left(\frac{T_{fast} + 273.15}{\theta}\right)^{0.286q_v}$$

$$T_{LCL} = 55 + \frac{2840}{3.5\ln\left(T_{fast} + 273.15\right) - \ln\left(e\right) - 4.805}$$

$$e = 0.01 \cdot RH \cdot e_s \quad q_v = 62.2\frac{e}{p} \quad e_s = 6.112^{\left[17.67(T-273.15)/243.5+(T-273.15)\right]} \quad T_{LCL} = 55 + \frac{2840}{3.5\ln\left(T\right) - \ln\left(e\right) - 4.805} \quad T_d = \frac{257.14\gamma}{18.678 - \gamma}$$

$$\gamma = \ln(0.01 \cdot RH)^{\left(\frac{18.678-T}{234.5} + \frac{T}{257.14+T}\right)}$$

Due to a hole in the pressure tube underneath the CoMeT-2 vehicle, it was found to have erroneously low air pressure measurements when the vehicle was in motion during LAPSE-RATE. To correct this error, observations from times when CoMeT-1 and CoMeT-2 were in motion and in close proximity were used to evaluate the level of inaccuracy of

the CoMeT-2 measurement. Here "close proximity" was defined as any observations within 25 meters of the same point, measured within 90 seconds of one another. The observations with the smallest distance between them were used, and duplicates were removed such that an observation from either vehicle was not used twice. The pressure difference and CoMeT-2 anemometer speed were then aligned with those from CoMeT-1 using a 2nd order polynomial. Anemometer speed was used instead of vehicle speed because vehicle speed was often a multiple of five, which made

it difficult to compute an accurate fit. The polynomial fit was used to calculate a pressure correction for all CoMeT-2 data obtained when the vehicle was in motion and the anemometer speed was greater than 10 m/s. Other variables calculated using pressure (e.g. $T_d$, $q_v$, $\theta$, $\theta_e$, and $\theta_v$) were recalculated using the corrected pressure.

Evaluation of data collected during the 14 July intercomparison along with an intercomparison conducted on 19 July

revealed an approximately constant bias in slow temperature in the CoMeT-1 data. The magnitude of this bias was approximated through minimization of the root mean square error across the intercomparison data sets and analysis of the adjusted time series. The result was a -0.6 K correction applied to all CoMeT-1 slow temperature data.

## 5 Data Availability and File Structure

The data files from the LAPSE-RATE project are generally being archived under a LAPSE-RATE community
established at the Zenodo data archive (https://zenodo.org/communities/lapse-rate/). From here, LAPSE-RATE
observations are available for public download and use. Contributors were encouraged to provide files in NetCDF
format, with self-describing metadata provided to the user inside the NetCDF file. To make it possible for scientists
to cite LAPSE-RATE data in their publications, the organizers of the campaign recognized the value of Digital Object
Identifiers (DOIs). DOIs were automatically generated by the Zenodo archive at the data version and product level.
Data from the different sources described above are posted as individual datastreams on the archive, with each of the
platforms described in the previous section having their own DOI. It is important to note that each platform may have
several different levels of data available. Therefore, data products with different levels of processing and quality
control may be provided with separate DOIs. This means the files and data described in this publication are spread
across a variety of DOIs, and that additional DOIs could be created in the future that include LAPSE-RATE data, as
additional data products are developed.

As of the writing of this manuscript, the CU MURC dataset (de Boer et al., 2020d) is available at Zenodo.org
(https://zenodo.org/record/3814765#.XrSRdS-z1TY) under DOI (http://doi.org/10.5281/zenodo.3814765). Data
from the NSSL MM includes two versions (Waugh, 2020). The original version contained files with incorrect quality
control (QC) flags. While the core data are correct, the QC flags can be useful for determining specific areas of interest
or problems. After identifying this issue, the files were reprocessed to include the correct QC flags and were uploaded
to the archive as version 2. Users should use version 2, which is available at Zenodo.org
(https://zenodo.org/record/3738175#.XrNLkC-z1TY) under DOI 10.5281/zenodo.37175. Finally, the UNL CoMeT
datasets (Houston and Erwin, 2020) also include two versions, and users are encouraged to use version 2 which
includes corrected GPS data for the vehicle locations. These data are also available at Zenodo.org
(https://zenodo.org/record/3838724#.XvOMGi2z1TZ) under DOI 10.5281/zenodo.3838724.

## 6 Summary

This manuscript provides an overview of data collected by three types of mobile surface systems during the 2018
LAPSE-RATE campaign. These included the University of Colorado MURC, the NOAA National Severe Storms
Laboratory Mobile Mesonet, and two University of Nebraska CoMeT vehicles. In combination, these vehicles
collected over 140 hours of meteorological data in the San Luis Valley of Colorado between 14-20 July, 2018. Data
from these vehicles are available for public download from zenodo.org, and the previous sections document processing
conducted on this dataset before publication, as well as information on the expected accuracy of the sensors deployed
on these systems. The primary focus of the LAPSE-RATE campaign was to collect data from a fleet of unmanned
aerial vehicles and surface in-situ and remote-sensing systems, and to combine those data with high-resolution
numerical simulations to gain understanding on boundary layer processes and phenomena. The primary measurement
objectives of the vehicles discussed in the current manuscript are shared above, along with the locations of operation
of each throughout the campaign.






**Author Contributions.** GB planned the LAPSE-RATE field campaign, constructed this manuscript, and conducted data processing of the CU MURC data. S.B., C.D. and B.A. contributed to the collection of the MURC data and were deployed to the field during LAPSE-RATE. A.E., W.S. and A.H. contributed to the collection of the UNL CoMeT data, processed and quality controlled these data, and are the primary points of contact for this dataset. They

additionally helped with the writing of this manuscript. S.W. was solely responsible for collection of the NSSL MM data and subsequent quality control of the resulting dataset. Additionally, he contributed to the writing of this manuscript.

**Competing Interests.** The authors claim no competing interests.


**Acknowledgements**. General support for salary and overhead associated with the collection of these datasets was provided by the NOAA Physical Sciences Division and the University of Colorado's Integrated Remote and In-Situ Sensing (IRISS) grand challenge project. We would additionally like to recognize financial support for student participation and travel from the National Science Foundation (NSF AGS 1807199) and the US Department of Energy

(DE-SC0018985). General support for the LAPSE-RATE campaign was provided by the International Society for Atmospheric Research using Remotely-piloted Aircraft (ISARRA). CoMeT-1 was funded through a grant from the Air Force Office of Scientific Research Defense University Research Instrumentation Program (FA2386-14-1-3010). CoMeT-2 was funded through an equipment allocation included in the NSF Research Infrastructure Improvement Program: Track-2 Focused EPSCoR Collaborations award (OIA-1539070). CoMeT-3 was funded through an

equipment allocation included in the NSF TORUS award (AGS-1824649). Funding for the NSSL MM and travel was provided for through internal NSSL funds, with sounding expendables donated by Oklahoma State University.

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



**Table 1:** Sensor specifications for the CU MURC

| Instrument Name | Observation | Range, Accuracy | Response time |
|---|---|---|---|
| Gill MetPak Pro | Air Temperature | -35-+70°C, ~±0.1°C | Not listed |
| | Relative Humidity | 0-100%, ±0.8% at 23 °C | Not listed |
| | Barometric Pressure | 600-1100 hPa, ±0.5 hPa | Not listed |
| | Wind Speed | 0-60 m/s, ± 2 % at 12 m/s | Not listed |
| | Wind Direction | 0-359°, ± 3° at 12 m/s | Not listed |
| Gill WindMaster 3D Sonic Anemometer | Wind Speed | 0-50 m/s, <1.5% RMS at 12 m/s | Not listed |
| | Wind Direction | 0-359°, ± 2° at 12 m/s | Not listed |
| | Sonic Temperature | -40-+70°C, < ± 0.5% at 20 °C | Not listed |
| RM Young Wind Monitor 05103 | Wind speed | 0-100 m/s, ± 0.3 m/s | 2.7 m |



**Table 2:** Sensor specifications for the NSSL MM

| Instrument Name | Observation | Range, Accuracy | Response time |
|---|---|---|---|
| Vaisala HMP155A | Air Temperature (slow) | -80-+60°C, ~±0.1°C | Not listed |
| | Relative Humidity | 0-100%, ±1% | 63% in 20s |
| Campbell Scientific T109SS | Air Temperature (fast) | -40-+70°C, ±0.6° | 7.5 s w/ 3m/s flow |
| RM Young Wind Monitor 05103 | Wind speed | 0-100 m/s, ± 0.3 m/s | 2.7 m |
| | Wind Direction | 0-360°, ± 3° | 1.3 m |
| Vaisala PTB210 | Barometric Pressure | 500-1100 mb, ±0.15 mb | n/a |
| KVH C100 Fluxgate | Magnetic Heading | 0-360°,.±0.16° | n/a |
| Garmin 19X HVS | Lat/Lon/Alt/Heading/Speed | n/a | n/a |






**Table 3:** Sensor specifications for the UNL CoMeT vehicles

| Instrument Name | Observation | Range, Accuracy | Response time |
|---|---|---|---|
| Vaisala HMP155A | Air Temperature (slow) | -80-+60°C, ~±0.1°C | Not listed |
| | Relative Humidity | 0-100%, ±1% | 63% in 20s |
| Campbell Scientific T109 SS | Air Temperature (fast) | -40-+70°C, ±0.6° | 7.5 s w/ 3m/s flow |
| RM Young Wind Monitor 05103 | Wind speed | 0-100 m/s, ± 0.3 m/s | 2.7 m |
| | Wind Direction | 0-360°, ± 3° | 1.3 m |
| Vaisala PTB210 | Barometric Pressure | 500-1100 mb, ±0.15 mb | n/a |
| KVH C100 Fluxgate | Magnetic Heading | 0-360°,.±0.16° | n/a |
| Garmin 19X HVS | Lat/Lon/Alt/Heading/Speed | n/a | n/a |

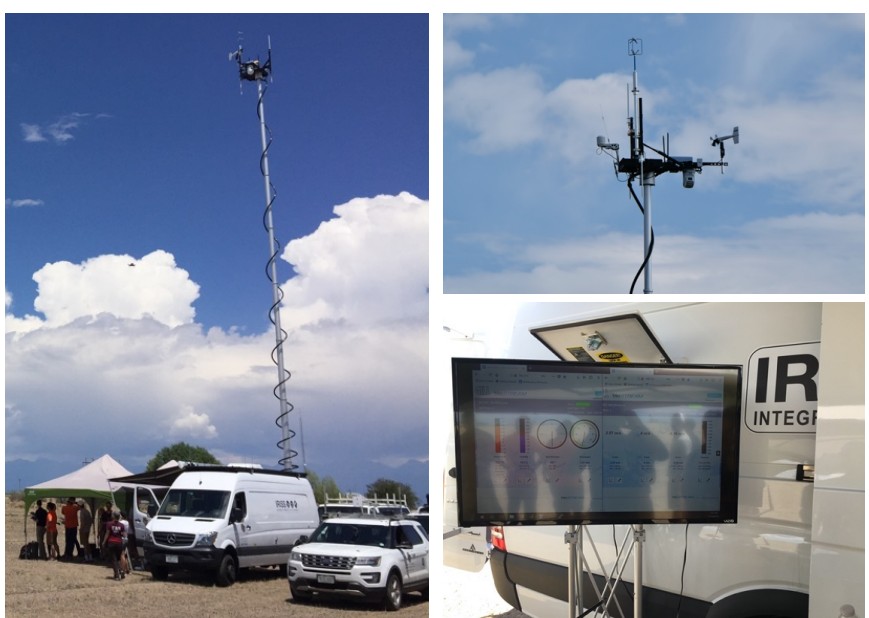

**Figure 1:** The CU MURC vehicle, with mast extended (left), as deployed during LAPSE-RATE. The right hand panels show the instrument cluster mounted on the top of the MURC mast (top) and the MURC real-time data display (bottom).

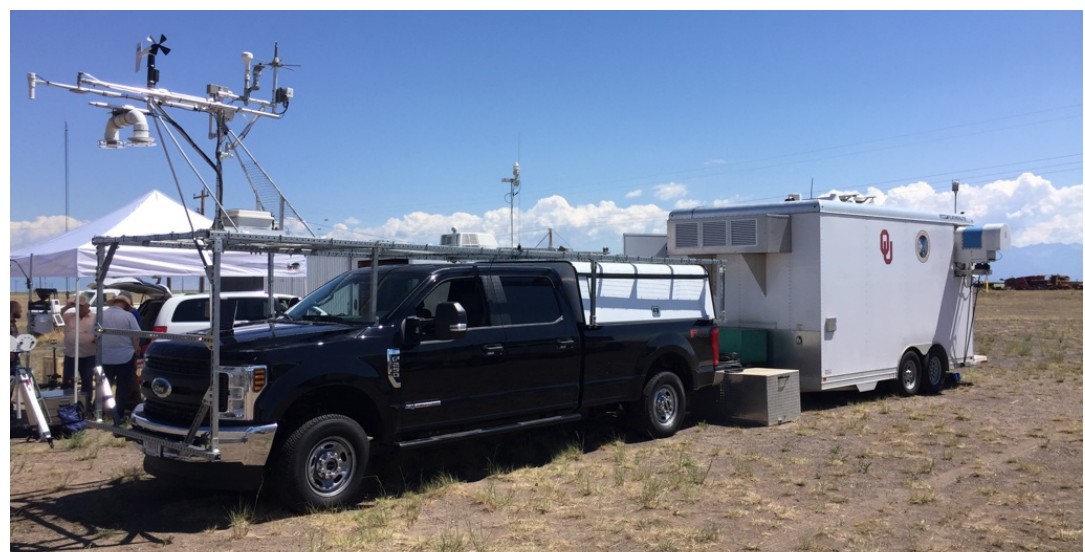

**Figure 2:** The NSSL MM as deployed during the LAPSE-RATE project. The trailer is the University of Oklahoma
CLAMPS system (see Bell et al., 2020 in this special issue).

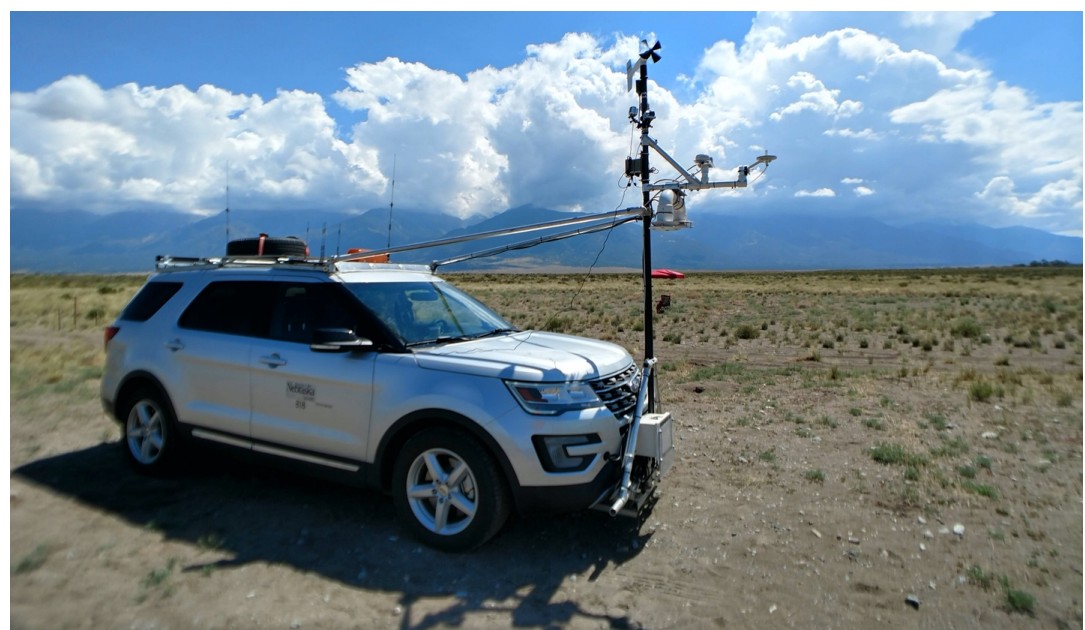

**Figure 3:** The UNL CoMeT vehicles, as deployed during LAPSE-RATE.


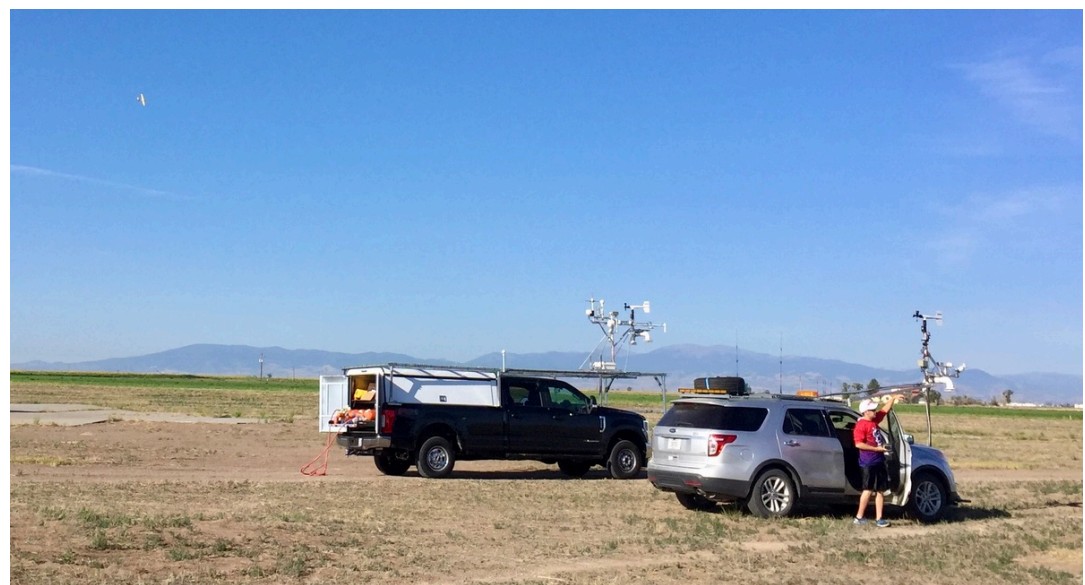

**Figure 4:** The NSSL MM and UNL CoMeT deployed side-by-side for an on-site intercomparison. The CU MURC was also located on site, but out of the photograph, and a CU TTwistor UAS (see de Boer et al., 2020c, in this special issue) flies in the background.




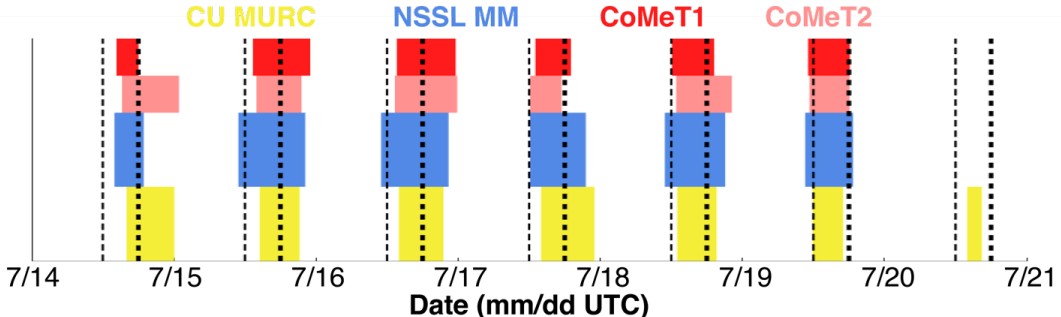

**Figure 5:** Illustration of the uptime for the mobile surface vehicles deployed during LAPSE-RATE. Shown is the update for the CU MURC (yellow), the NSSL MM (blue) and the UNL CoMeTs (red and pink). The thin dashed black lines indicate 0600 local time (Mountain Daylight Time), while the bold dashed black lines represent 1200 local time for each day.


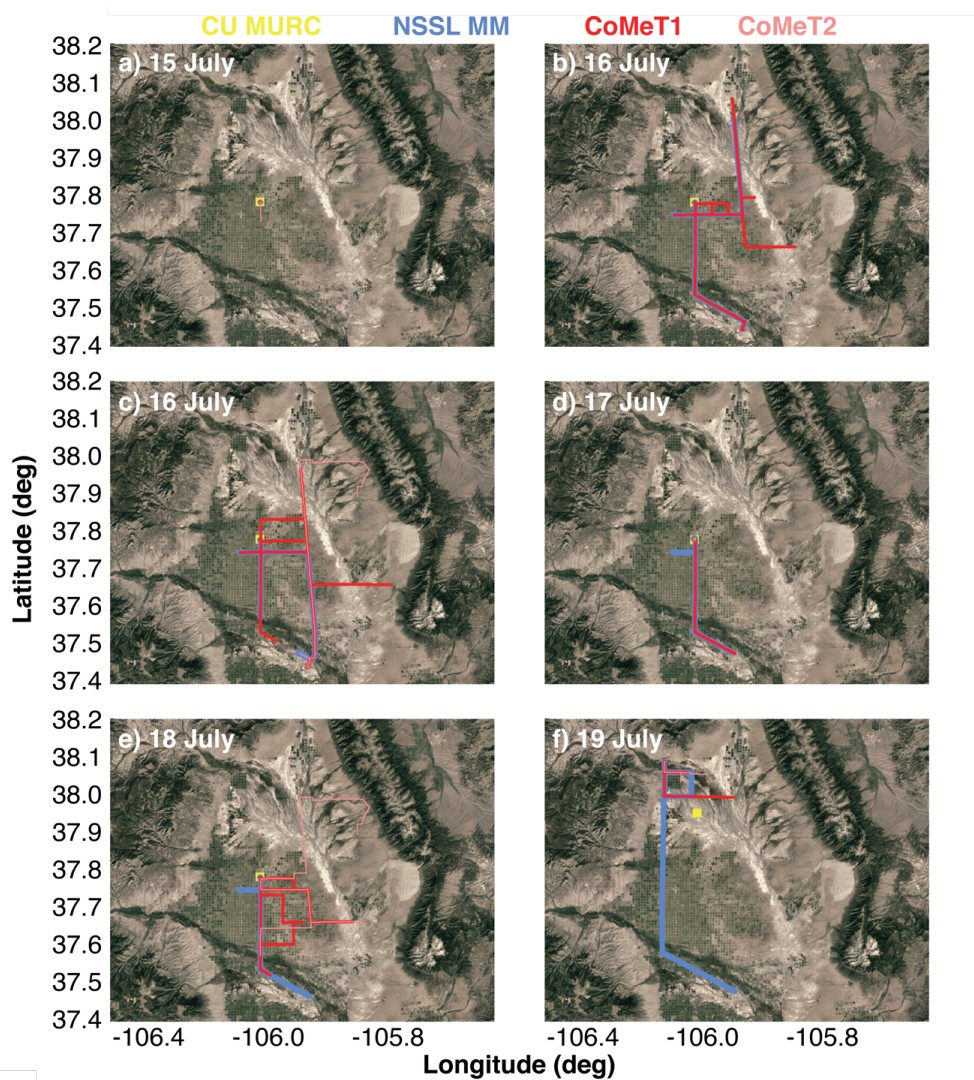

**Figure 6:** Position data for the different platforms over the length of the campaign, broken down on a day-by-day basis. The CU MURC positions are indicated by the yellow square, while the NSSL mobile mesonet (blue) and UNL CoMeTs (red and pink) mobile datasets are shown by the lines. Background maps are from © Google through their API.

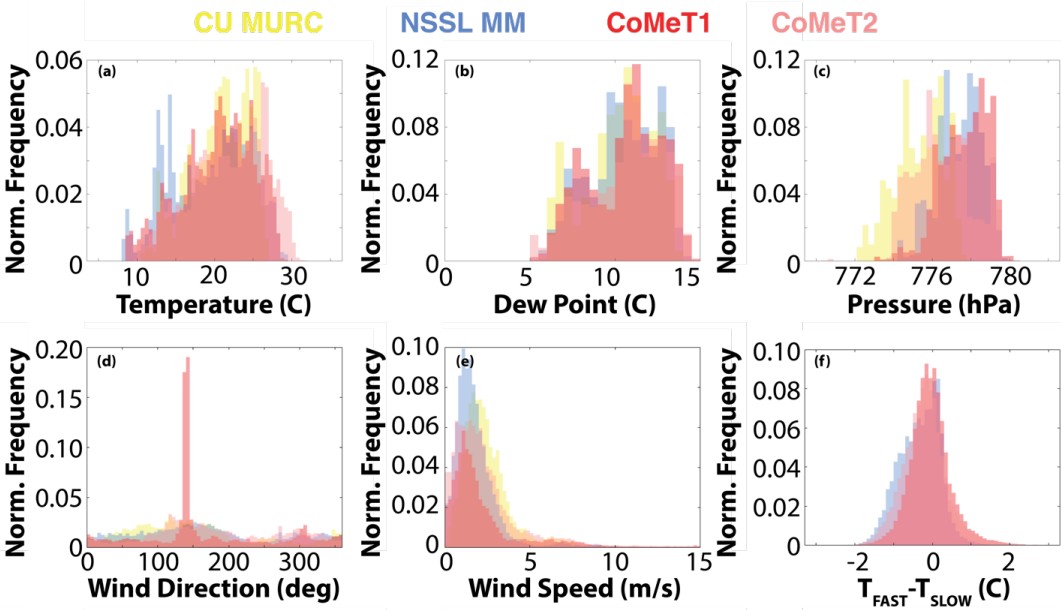


**Figure 7:** Distributions of 1-minute averages of data collected by the three platforms. Included are (a-f) distributions of air temperature, dew point temperature, pressure, wind direction, wind speed, and the difference between the fast and slow temperature sensors (where applicable). For all figures, CU MURC data are represented in yellow, NSSL MM data are represented in blue, and UNL CoMeT data are represented in red and pink.


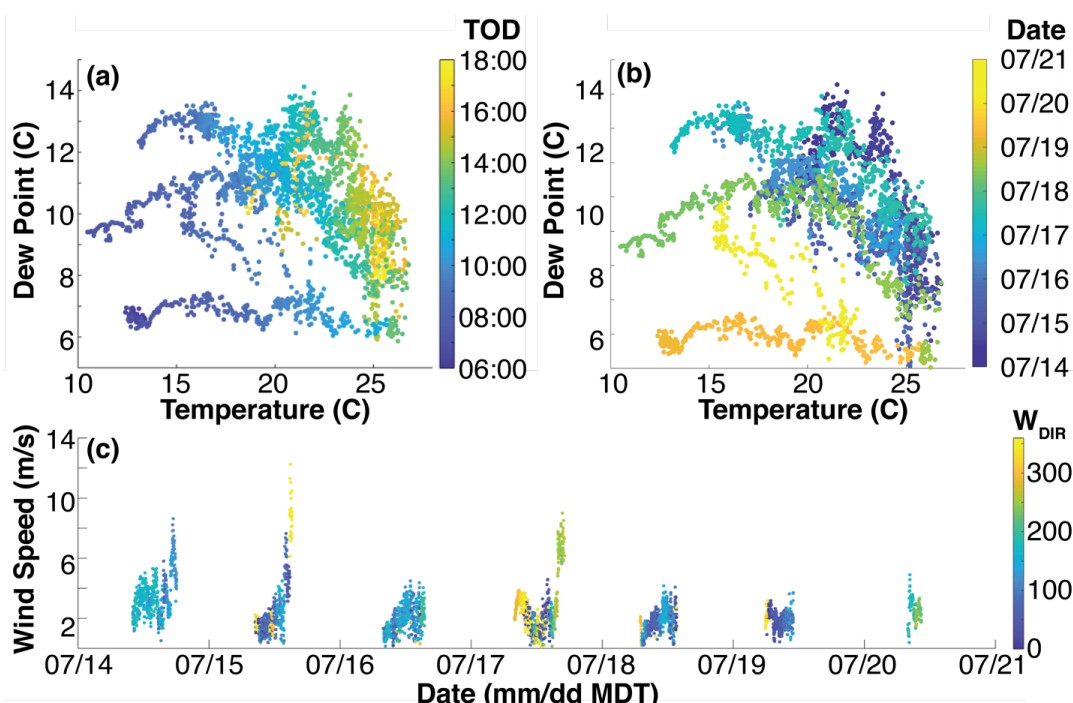

**Figure 8:** Panels illustrating the temporal variability of the temperature (a,b), dew point temperature (a,b) and wind data (c) collected by the CU MURC. Panel a shows diurnal variability, while panel b shows the variability by date.