# Peer review of "Measurements from mobile surface vehicles during LAPSE-RATE"

_Earth System Science Data, 2020_

## Referee Comment (RC1) · Gijs de Boer et al. · 25 Sep 2020

Review ESSD-2020-173

LAPSE data nicely organised on Zenodo, easy to access and download files specific to this manuscript.

Line 213: "50-foot mast" Reading a tower height in US units here seems a bit strange after description of most measurements on most vehicles in metric units up to this point.

Line. 390: "close proximity" Many readers / data users will find the time periods of proximity useful for data intercomparison assessments. As implied by Figure 4 (and perhaps also by Figure 7 but harder to distinguish), vehicle proximity not only of CoMeT-1 to CoMeT-2 but also between NSSL and UNL or among all three could have occurred multiple times per day? I do not see these periods identified in the CoMeT-1 data nor flagged in the NSSL data. User would need to find periods when GPS lat, lon and UTC coincide among two, three or four platforms? These authors will know data quality and utility better than outside reviewers; would an explicit summary of these proximity opportunities represent a useful addition? Eventually, direct intercomparison periods among sUAS and ground-based systems will drive an overall all intercomparison opportunity chart / graphic?

Funding for three CoMeT vehicles mentioned in acknowledgements but only two vehicles used in the deployment described here?

Table 2 & Table 3: response time for RM Young propeller-vane anemometer shown in units of "m" in both tables. But both of those tables also use 'm' as a length unit, e.g. m/s. Response time of anemometers in 'm' = minutes? E.g. 2.7 minutes for speed, 1.3 minutes for direction? That seems too slow? Check these units? Do not use "m" to designate both length and time?

---

## Referee Comment (RC2) · Anonymous Referee #2 · 16 Oct 2020

The title of the manuscript is very attractive and this work is also very meaningful. But after reading the entire article, I was still a little disappointed. For articles published in ESSD, the production of data sets and their quality evaluation are the most important. However, the depth of the current version of the article is not enough. The full text looks like a report, with too many lists and simple descriptions, not a research paper. The work is well done, but the organization and presentation of the article is not enough.

Major comments: 1. The observation period (14 and 20 July 2018) of the data is too short. It is difficult to say that this data can have too much contribution to scientific researchers around the world. But the research methods (mobile surface vehicles) are very meaningful. 2. The observation items of the data are too conventional, basically meteorological data (temperature, wind speed, temperature, etc.), without special data.

[Figure]

The article's comparison of these data is also relatively superficial. 3. If there is an introduction about the route setting, the structure of the entire equipment, and the cost, it may provide a more valuable reference for related scientists. 4. Data processing and quality control should not be considered as innovations of this article, but the article uses a larger amount of space. What do you want to express? 5. The biggest problem with mobile observation maybe its representation of time and space. Has the author elaborated this in the manuscript? How the author chooses the route and the sampling time? 6. The introduction of observation items and instruments (Table1-3) can be integrated into a table so that everyone can understand the system more directly.

In summary, I don't think the current version is suitable for publication on ESSD.

---

## Author Response (AR1)

This document provides a response to the comments raised by the two reviewers who have provided interactive comments on manuscript essd-2020-173 ("Measurements from mobile surface vehicles during LAPSE-RATE"). The reviewer comments in included in **black** and author responses are included in **red**.

General comments from the authors: We appreciate the time provided by the two reviewers and the editorial team and have done our best to address the comments provided. Additionally, we have updated references to meet ESSD formatting requirements and include the most up-to-date information. Finally, in review of this manuscript we noticed an error with one of the figures (7d), where for CoMeT-1, vehicle heading had mistakenly been plotted instead of wind direction. Therefore, we have updated that figure, resulting in the removal of the large "spike" at around 140 degrees that was present in the previous version. Direct responses to the reviewer comments are provided below.

**Reviewer 1:**

Review ESSD-2020-173

LAPSE data nicely organised on Zenodo, easy to access and download files specific to this manuscript.

Thank you for this positive comment. We are also impressed with Zenodo as an archive and will likely use it for publication of future results!

Line 213: "50-foot mast" Reading a tower height in US units here seems a bit strange after description of most measurements on most vehicles in metric units up to this point.

Thank you for pointing this out – we have updated the text to read "15.2 m mast"

Line. 390: "close proximity" Many readers / data users will find the time periods of proximity useful for data intercomparison assessments. As implied by Figure 4 (and perhaps also by Figure 7 but harder to distinguish), vehicle proximity not only of CoMeT-1 to CoMeT-2 but also between NSSL and UNL or among all three could have occurred multiple times per day? I do not see these periods identified in the CoMeT-1 data nor flagged in the NSSL data. User would need to find periods when GPS lat, lon and UTC coincide among two, three or four platforms? These authors will know data quality and utility better than outside reviewers; would an explicit summary of these proximity opportunities represent a useful addition? Eventually, direct intercomparison periods among sUAS and ground-based systems will drive an overall all intercomparison opportunity chart / graphic?

The reviewer is correct that there is no "proximity flag" available in the data that are posted to Zenodo. While we understand the potential usefulness of such a flag, it is also important to understand that these vehicles are not always deployed together and therefore implementing such a flag is not practical from the perspective of the standard processing associated with these systems.

Given that all three vehicle files report GPS position, it should not be too difficult for a user to establish when the vehicles were in "close proximity". At the recommendation of the reviewer (and, to some extent Reviewer #2), we have added a paragraph that offers an overview of direct comparison data from the time periods where the vehicles were close together (< 300 m apart), and added a figure (figure 9) that provides a direct comparison of the measurements during these times.

Funding for three CoMeT vehicles mentioned in acknowledgements but only two vehicles used in the deployment described here?

Thank you for pointing this out – it was a copy and paste error, and the reference to CoMeT-3 has been removed from the acknowledgments.

Table 2 & Table 3: response time for RM Young propeller-vane anemometer shown in units of "m" in both tables. But both of those tables also use 'm' as a length unit, e.g. m/s. Response time of anemometers in 'm' = minutes? E.g. 2.7 minutes for speed, 1.3 minutes for direction? That seems too slow? Check these units? Do not use "m" to designate both length and time?

The response "time" of wind sensors for cup and propeller systems is in fact measured in meters, not seconds. It's not a response time so much as a distance constant. Literally the length of fluid flow required to result in a 1/e response to a change in the observed winds. We have updated the entry into the tables (now tables 1 and 2, in response to comments from reviewer 2) to include the comment "distance constant".

**Reviewer 2:**

The title of the manuscript is very attractive and this work is also very meaningful. But after reading the entire article, I was still a little disappointed. For articles published in ESSD, the production of data sets and their quality evaluation are the most important. However, the depth of the current version of the article is not enough. The full text looks like a report, with too many lists and simple descriptions, not a research paper. The work is well done, but the organization and presentation of the article is not enough.

Major comments:

1. The observation period (14 and 20 July 2018) of the data is too short. It is difficult to say that this data can have too much contribution to scientific researchers around the world. But the research methods (mobile surface vehicles) are very meaningful.

While we respect the reviewer's opinion, we disagree with this assessment. There are already published and in-progress studies conducted leveraging the LAPSE-RATE data, including the data collected from these vehicles. It is important to remember that these measurements are part of a bigger campaign that featured over 50 UAS platforms, and a variety of other surface-based remote sensors. The papers that have been prepared include a UAS measurement intercomparison study (Barbieri et al., 2019), advancement of UAS capabilities (Islam et al., 2019), a study to evaluate

the ability of UAS to track coherent atmospheric structures (Nolan et al., 2018), in addition to two in-preparation articles evaluating the influence of assimilated UAS observations on prediction of weather in complex terrain (Jensen et al., 2020a and 2020b), and an in-preparation paper evaluating the structure and intensity of cold-air drainage from the Saguache Canyon (Bailey et al., 2020).

2. The observation items of the data are too conventional, basically meteorological data (temperature, wind speed, temperature, etc.), without special data. The article's comparison of these data is also relatively superficial.

We're not sure what the reviewer means by "special data". We believe that temperature, wind speed, humidity and other standard meteorological quantities represent the backbone of much of our understanding of the atmosphere. We do not believe that ESSD is in place only to support the publication of "special" data, but rather that the publication is meant to support the documentation and publication of all datasets related to Earth Science.

Regarding the comparison of the data, we included Figure 7 to provide some information on how the different measurements compare. We realize that this is not an in-depth comparison, though each of these systems undergoes routine calibrations at their home facilities. Additionally, we have added a second figure (figure 9) from time periods where the different vehicles were in close proximity (< 300 m apart), and added some text describing this figure. Note that we believe that the observations we currently have do not support an in-depth intercomparison between these vehicles (that wasn't the point of this field campaign), and that doing that well would require a new campaign (and a lot of data analysis that extends beyond the reaches of an ESSD article).

3. If there is an introduction about the route setting, the structure of the entire equipment, and the cost, it may provide a more valuable reference for related scientists.

We're not sure that we understand this comment. What is meant by the "route setting"? Is this in reference to determining where the vehicles would be positioned on a given date? If so, we believe that this is captured in section 3 (lines 168-205) and the references provided to offer a broader overview of the LAPSE-RATE campaign (de Boer et al., 2020a and 2020b). Regarding the "structure of the entire equipment and the cost", we believe that section 2 and references therein provide adequate descriptions of the systems. While the cost is not specifically included, we don't necessarily believe that this is relevant for documentation of the collected dataset (primary objective of ESSD), and, frankly doesn't seem to be something that needs to be published. Given the rapid change in equipment and instrumentation pricing, along with the massive differences in vehicle costs, there is no clear way to provide a regionally and temporally accurate cost estimate that could be used in any meaningful way by the reader.

4. Data processing and quality control should not be considered as innovations of this article, but the article uses a larger amount of space. What do you want to express?

ESSD articles, in general, are not the place to publish "innovations". Those are more suitably published in either technology-centric journals (e.g. AMT, J. Tech) or scientific journals (ACP, AMS, AGU, Springer journals). The text provided in the current article is meant to provide the reader with background on the processes employed to collect and process this dataset. We believe

that we have done a reasonable job with offering the reader insight into what calculations are performed, what quality control is applied, and what sensors are used. For many of the applied techniques, we provide references the offer details on the earlier innovations supporting the development and collection of this dataset.

5. The biggest problem with mobile observation maybe its representation of time and space. Has the author elaborated this in the manuscript? How the author chooses the route and the sampling time?

It is true that connecting the temporal and spatial variability observed using the mobile observing platforms can pose challenges. The extent to which these challenges need to be overcome are specific to the scientific questions to be answered and the phenomena to be observed. As an example, the spatial and temporal variability in these observations is actually quite useful for trying to evaluate the physical characteristics and drivers of something like a valley drainage flow. We did not dedicate time in the current manuscript on this topic given that this would require an in depth analysis to understand the considerations for answering a given scientific question of interest, which is broadly beyond the scope of an ESSD publication.

With regard to the selected routes and sampling times for the current dataset, the rationales supporting the deployment locations of each platform were described in Section 3 and the cited publications therein. The times selected were meant to align with scientific objectives of the LAPSE-RATE campaign (e.g. detecting early morning drainage flows, capturing the boundary layer evolution, assessing the initiation and development of convection through the morning into the early afternoon).

6. The introduction of observation items and instruments (Table1-3) can be integrated into a table so that everyone can understand the system more directly.

We're not sure that we understand the comment. If we read it directly, the reviewer is requesting that the information provided in table form be put into a table, which doesn't make sense. Possibly the suggestion is to combine these items into a single table, though the current breakdown is meant to offer individual tables for the three different types of mobile systems deployed during LAPSE-RATE. Given that the NSSL MM and UNL CoMeT systems have identical instrumentation, we have combined tables 2 and 3 into a single table 2. Beyond this, we would prefer to keep the MURC table separate from the other two. We apologize if we misunderstood the reviewer's comment.

In summary, I don't think the current version is suitable for publication on ESSD.

Nevertheless, we appreciate the time provided by the reviewer in commenting on this manuscript. We strongly believe that this article is a good fit for ESSD, and would be happy to hear the editorial staff's opinion on whether additional components should be included prior to publication.

Cited References:

Barbieri, L.K., S.T. Kral, S.C.C. Bailey, A.E. Frazier, J.D. Jacob,D. Brus, P.B. Chilson, C. Crick, J. Elston, H. Foroutan, J. González-Rocha, B.R. Greene, M.I. Guzman, A.L. Houston, A. Islam,O. Kemppinen, E.A. Pillar-Little, J. Reuder, S.D. Ross, M. Sama, D.G. Schmale III, T.J. Schuyler, S. Smith, S. Waugh, A. Doddi, D. Lawrence, C. Dixon, S. Borenstein, and G. de Boer (2019): Intercomparison of small unmanned aircraft system (sUAS) measurements for atmospheric science during the LAPSE-RATE campaign, *Sensors*, 19, 2179, https://doi.org/10.3390/s19092179

de Boer, G., Diehl, C., Jacob, J., Houston, A., Smith, S.W., Chilson, P., Schmale III, D.G., Intrieri, J., Pinto, J., Elston, J., Brus, D., Kemppinen, O., Clark, A., Lawrence, D., Bailey, S.C.C., Sama, M.P., Frazier, A., Crick, C., Natalie, V., Pillar-Little, E.A., Klein, P., Waugh, S., Lundquist, J.K., Barbieri, L., Kral, S.T., Jensen, A.A., Dixon, C., Borenstein, S., Hesselius, D., Human, K., Hall, P., Argrow, B., Thornberry, T., Wright, R. and Kelly, J.T.: Development of community, capabilities and understanding through unmanned aircraft-based atmospheric research: The LAPSE-RATE campaign, *Bull. Amer. Meteor. Soc.*, **101**, E684-E699, https://doi.org/10.1175/BAMS-D-19-0050.1, 2020a.

de Boer, G., Houston, A., Jacob, J., Chilson, P. B., Smith, S. W., Argrow, B., Lawrence, D., Elston, J., Brus, D., Kemppinen, O., Klein, P., Lundquist, J. K., Waugh, S., Bailey, S. C. C., Frazier, A., Sama, M. P., Crick, C., Schmale III, D., Pinto, J., Pillar-Little, E. A., Natalie, V., and Jensen, A.: Data Generated During the 2018 LAPSE-RATE Campaign: An Introduction and Overview, Earth Syst. Sci. Data, https://doi.org/10.5194/essd-2020-98, accepted for publication, 2020b.

Islam, A., Houston, A.L., Shankar, A. and Detweiler, C.: Design and Evaluation of Sensor Housing for Boundary Layer Profiling Using Multirotors. *Sensors*, **19**, 2481, doi: 10.3390/s19112481, 2019.

Jensen, A., and co-authors: Assimilation of a coordinated fleet of unmanned aircraft systems observations in complex terrain. Part I: EnKF system design and preliminary assessment, *Mon. Wea. Rev.*, submitted, 2020.

Nolan, P., J. Pinto, J. Gonzalez-Rocha, A. Jensen, C.N. Vezzi, S.C.C. Bailey, G. de Boer, C. Diehl, R. Laurence III, C.W. Powers, H. Foroutan, S.D. Ross and D.G. Schmale III: Coordinated unmanned aircraft system (UAS) and ground-based weather measurements to predict lagrangian coherent structures (LCSs), *Sensors*, 18, 4448, https://doi.org/10.3390/s18124448

[revised manuscript text omitted]

**Formatted** ... [2]

**Moved down [1]:** A. L.

**Moved (insertion) [1]**

**Moved down [2]:** R. J.

**Moved (insertion) [2]**

**Moved down [3]:** T. W.

**Moved (insertion) [3]**

**Moved down [4]:** C. L.

**Moved (insertion) [4]**

**Moved down [5]:** A. L.

**Moved (insertion) [5]**

LAPSE-RATE field campaign, Earth Syst. Sci. Data Discuss., https://doi.org/10.5194/essd-2020-194, in review, 2020.

Pinto, J. O., Jensen, A. A., Jiménez, P. A., Hertneky, T., Muñoz-Esparza, D., Dumont, A., and Steiner, M.: Realtime WRF LES Simulations to Support UAS Flight Planning and Operations During 2018 LAPSE-RATE, Earth Syst. Sci. Data Discuss., https://doi.org/10.5194/essd-2020-242, in review, 2020.

Rasmussen, E. N., Straka, J. M., Davies-Jones, R., Doswell III, C. A., Carr, F. H., Eilts, M. D. and MacGorman, D. R.: Verification of the Origins of Rotation in Tornadoes Experiment: VORTEX. *Bull. Amer. Meteor. Soc.*, **75**, 995–1006, https://doi.org/10.1175/1520-0477(1994)075,0995:VOTOOR.2.0.CO;2, 1994.

Richardson, S. J., Frederickson, S. E., Brock, F. V. and Brotzge, J. A.: Combination temperature and relative humidity probes: Avoiding large air temperature errors and associated relative humidity errors. Preprints, 10th Symp. On Meteorological Observations and Instrumentation, Phoenix, AZ, Amer. Meteor. Soc., 278–283, 1998.

Riganti, C.J. and Houston, A.L.: Rear-Flank Outflow Dynamics and Thermodynamics in the 10 June 2010 Last Chance, Colorado, Supercell. *Mon. Wea. Rev.*, **145**, 2487–2504, https://doi.org/10.1175/MWR-D-16-0128.1, 2017.

Straka, J. M., Rasmussen, E. N. and Fredrickson, S. E.: A mobile mesonet for finescale meteorological observations. *J. Atmos. Oceanic Technol.*, **13**, 921–936, https://doi.org/10.1175/1520-0426(1996)013<0921:AMMFFM>2.0.CO;2, 1996.

Tropea, C., Yarin, A.L. and Foss, J.F. (Ed.). *Springer Handbook of Experimental Fluid Mechanics*, Berlin, Springer, https://doi.org/10.1007/978-3-540-30299-5, 2007.

Waugh, S.: The "U-tube": An improved aspirate temperature system for mobile meteorological observations, especially in severe weather. MS Thesis, Univ. of Oklahoma, 87 pp., 2012.

Waugh, S. and Frederickson, S. E.: An improved aspirated temperature system for mobile meteorological observations, especially in severe weather. 25th Conf. on Severe Local Storms, Denver, CO, Amer. Meteor. Soc., P 5.2. [Available online at https://ams.confex.com/ams/25SLS/techprogram/paper_176205.htm.], 2010.

Waugh, S.: National Severe Storms Laboratory Mobile Mesonet data files from Lapse-Rate [Data set]. Zenodo. http://doi.org/10.5281/zenodo.3738175, 2020.

**Table 1:** Sensor specifications for the CU MURC

| Instrument Name | Observation | Range, Accuracy | Response time |
|---|---|---|---|
| Gill MetPak Pro | Air Temperature | -35-+70°C, ~±0.1°C | Not listed |
| | Relative Humidity | 0-100%, ±0.8% at 23 °C | Not listed |
| | Barometric Pressure | 600-1100 hPa, ±0.5 hPa | Not listed |
| | Wind Speed | 0-60 m/s, ± 2 % at 12 m/s | Not listed |
| | Wind Direction | 0-359°, ± 3° at 12 m/s | Not listed |
| Gill WindMaster 3D Sonic Anemometer | Wind Speed | 0-50 m/s, <1.5% RMS at 12 m/s | Not listed |
| | Wind Direction | 0-359°, ± 2° at 12 m/s | Not listed |
| | Sonic Temperature | -40-+70°C, < ± 0.5% at 20 °C | Not listed |
| RM Young Wind Monitor 05103 | Wind speed | 0-100 m/s, ± 0.3 m/s | 2.7 m [distance constant] |

925

**Table 2:** Sensor specifications for the NSSL MM and UNL CoMeT vehicles

| Instrument Name | Observation | Range, Accuracy | Response time |
|---|---|---|---|
| Vaisala HMP155A | Air Temperature (slow) | -80-+60°C, ~±0.1°C | Not listed |
| | Relative Humidity | 0-100%, ±1% | 63% in 20s |
| Campbell Scientific T109SS | Air Temperature (fast) | -40-+70°C, ±0.6° | 7.5 s w/ 3m/s flow |
| RM Young Wind Monitor 05103 | Wind speed | 0-100 m/s, ± 0.3 m/s | 2.7 m [distance constant] |
| | Wind Direction | 0-360°, ± 3° | 1.3 m [distance constant] |
| Vaisala PTB210 | Barometric Pressure | 500-1100 mb, ±0.15 mb | n/a |
| KVH C100 Fluxgate | Magnetic Heading | 0-360°,.±0.16° | n/a |
| Garmin 19X HVS | Lat/Lon/Alt/Heading/Speed | n/a | n/a |

930

¶

**Table 3:** Sensor specifications for the UNL CoMeT vehicles

935

[Figure]

**Figure 1:** The CU MURC vehicle, with mast extended (left), as deployed during LAPSE-RATE. The right hand panels show the instrument cluster mounted on the top of the MURC mast (top) and the MURC real-time data display (bottom).

940

[Figure]

**Figure 2:** The NSSL MM as deployed during the LAPSE-RATE project. The trailer is the University of Oklahoma CLAMPS system (see Bell et al., 2020 in this special issue).

945

[Figure]

**Figure 3:** The UNL CoMeT vehicles, as deployed during LAPSE-RATE.

[Figure]

**Figure 4:** The NSSL MM and UNL CoMeT deployed side-by-side for an on-site intercomparison. The CU MURC
was also located on site, but out of the photograph, and a CU TTwistor UAS (see de Boer et al., 2020c, in this special
issue) flies in the background.

950

[Figure]

**Figure 5:** Illustration of the uptime for the mobile surface vehicles deployed during LAPSE-RATE. Shown is the
update for the CU MURC (yellow), the NSSL MM (blue) and the UNL CoMeTs (red and pink). The thin dashed
black lines indicate 0600 local time (Mountain Daylight Time), while the bold dashed black lines represent 1200 local
time for each day.

[Figure]

960

**Figure 6:** Position data for the different platforms over the length of the campaign, broken down on a day-by-day basis. The CU MURC positions are indicated by the yellow square, while the NSSL mobile mesonet (blue) and UNL CoMeTs (red and pink) mobile datasets are shown by the lines. Background maps are from © Google through their API.

965

[Figure]

**Figure 7:** Distributions of 1-minute averages of data collected by the three platforms. Included are (a-f) distributions of air temperature, dew point temperature, pressure, wind direction, wind speed, and the difference between the fast and slow temperature sensors (where applicable). For all figures, CU MURC data are represented in yellow, NSSL MM data are represented in blue, and UNL CoMeT data are represented in red and pink.

[Figure]

**Figure 8:** Panels illustrating the temporal variability of the temperature (a,b), dew point temperature (a,b) and wind data (c) collected by the CU MURC. Panel a shows diurnal variability, while panel b shows the variability by date.

975

[Figure]

**Figure 9:** Vehicle-to-vehicle comparisons for time periods when vehicles were within 300 m of one another. Variables evaluated include (top to bottom) temperature (C), relative humidity (%), pressure (hPa), wind speed (m s$^{-1}$) and wind direction (deg). The top row includes labels to indicate which platform is on which axis, and these orientations are maintained through each column. For the temperature comparisons (top row), both the slow (dark dots) and fast (lighter dots) temperature sensors are evaluated.

980

| Page 15: [1] Deleted | Gijs de Boer | 10/27/20 11:41:00 AM |
|---|---|---|

| Page 16: [2] Formatted | Gijs de Boer | 10/27/20 11:43:00 AM |
|---|---|---|

Font: Times New Roman, 10 pt, Spanish

| Page 16: [2] Formatted | Gijs de Boer | 10/27/20 11:43:00 AM |
|---|---|---|

Font: Times New Roman, 10 pt, Spanish

| Page 16: [3] Deleted | Gijs de Boer | 10/27/20 11:44:00 AM |
|---|---|---|

| Page 16: [3] Deleted | Gijs de Boer | 10/27/20 11:44:00 AM |
|---|---|---|

| Page 16: [3] Deleted | Gijs de Boer | 10/27/20 11:44:00 AM |
|---|---|---|

| Page 16: [4] Deleted | Gijs de Boer | 10/27/20 12:25:00 PM |
|---|---|---|

| Page 16: [4] Deleted | Gijs de Boer | 10/27/20 12:25:00 PM |
|---|---|---|

| Page 16: [4] Deleted | Gijs de Boer | 10/27/20 12:25:00 PM |
|---|---|---|

| Page 16: [4] Deleted | Gijs de Boer | 10/27/20 12:25:00 PM |
|---|---|---|

| Page 16: [4] Deleted | Gijs de Boer | 10/27/20 12:25:00 PM |
|---|---|---|

| Page 16: [4] Deleted | Gijs de Boer | 10/27/20 12:25:00 PM |
|---|---|---|

| Page 16: [4] Deleted | Gijs de Boer | 10/27/20 12:25:00 PM |
|---|---|---|

| Page 16: [4] Deleted | Gijs de Boer | 10/27/20 12:25:00 PM |
|---|---|---|

| Page 16: [4] Deleted | Gijs de Boer | 10/27/20 12:25:00 PM |
|---|---|---|

| Page 16: [4] Deleted | Gijs de Boer | 10/27/20 12:25:00 PM |
|---|---|---|

| Page 16: [4] Deleted | Gijs de Boer | 10/27/20 12:25:00 PM |

| Page 16: [4] Deleted | Gijs de Boer | 10/27/20 12:25:00 PM |

| Page 16: [4] Deleted | Gijs de Boer | 10/27/20 12:25:00 PM |

| Page 16: [5] Deleted | Gijs de Boer | 10/27/20 11:50:00 AM |

| Page 16: [5] Deleted | Gijs de Boer | 10/27/20 11:50:00 AM |

| Page 16: [6] Deleted | Gijs de Boer | 10/27/20 11:50:00 AM |

| Page 16: [6] Deleted | Gijs de Boer | 10/27/20 11:50:00 AM |

| Page 16: [6] Deleted | Gijs de Boer | 10/27/20 11:50:00 AM |

| Page 16: [7] Deleted | Gijs de Boer | 10/27/20 11:50:00 AM |

| Page 16: [7] Deleted | Gijs de Boer | 10/27/20 11:50:00 AM |

| Page 16: [8] Deleted | Gijs de Boer | 10/27/20 12:27:00 PM |

| Page 16: [8] Deleted | Gijs de Boer | 10/27/20 12:27:00 PM |

| Page 16: [9] Deleted | Gijs de Boer | 10/27/20 11:45:00 AM |

| Page 16: [10] Deleted | Gijs de Boer | 10/27/20 12:29:00 PM |

| Page 16: [11] Deleted | Gijs de Boer | 10/27/20 12:34:00 PM |

| Page 16: [11] Deleted | Gijs de Boer | 10/27/20 12:34:00 PM |

| Page 16: [11] Deleted | Gijs de Boer | 10/27/20 12:34:00 PM |

| Page 16: [11] Deleted | Gijs de Boer | 10/27/20 12:34:00 PM |

| Page 16: [12] Deleted | Gijs de Boer | 10/27/20 12:35:00 PM |

| Page 16: [12] Deleted | Gijs de Boer | 10/27/20 12:35:00 PM |

| Page 16: [13] Deleted | Gijs de Boer | 10/27/20 11:45:00 AM |
| Page 16: [13] Deleted | Gijs de Boer | 10/27/20 11:45:00 AM |
| Page 16: [13] Deleted | Gijs de Boer | 10/27/20 11:45:00 AM |
| Page 17: [14] Deleted | Gijs de Boer | 10/27/20 11:46:00 AM |
| Page 17: [15] Deleted | Gijs de Boer | 10/27/20 11:46:00 AM |
| Page 17: [16] Deleted | Gijs de Boer | 10/27/20 12:38:00 PM |

| Page 17: [16] Deleted | Gijs de Boer | 10/27/20 12:38:00 PM |

| Page 17: [17] Deleted | Gijs de Boer | 10/27/20 12:39:00 PM |

| Page 17: [17] Deleted | Gijs de Boer | 10/27/20 12:39:00 PM |

| Page 17: [18] Deleted | Gijs de Boer | 10/27/20 12:39:00 PM |

| Page 17: [18] Deleted | Gijs de Boer | 10/27/20 12:39:00 PM |

| Page 17: [19] Formatted | Gijs de Boer | 10/28/20 2:29:00 PM |

Swedish

| Page 17: [20] Formatted | Gijs de Boer | 10/27/20 12:39:00 PM |

Swedish

| Page 17: [21] Change | Unknown | |

Field Code Changed

| Page 17: [22] Formatted | Gijs de Boer | 10/27/20 12:39:00 PM |

Swedish

| Page 17: [22] Formatted | Gijs de Boer | 10/27/20 12:39:00 PM |

Swedish

| Page 17: [23] Deleted | Gijs de Boer | 10/27/20 12:39:00 PM |

| Page 17: [23] Deleted | Gijs de Boer | 10/27/20 12:39:00 PM |
|---|---|---|

| Page 17: [24] Formatted | Gijs de Boer | 10/28/20 2:30:00 PM |
|---|---|---|

English (US)

| Page 17: [25] Deleted | Gijs de Boer | 10/27/20 12:40:00 PM |
|---|---|---|

| Page 17: [25] Deleted | Gijs de Boer | 10/27/20 12:40:00 PM |
|---|---|---|

| Page 17: [25] Deleted | Gijs de Boer | 10/27/20 12:40:00 PM |
|---|---|---|

| Page 17: [25] Deleted | Gijs de Boer | 10/27/20 12:40:00 PM |
|---|---|---|

| Page 17: [25] Deleted | Gijs de Boer | 10/27/20 12:40:00 PM |
|---|---|---|

| Page 17: [26] Deleted | Gijs de Boer | 10/27/20 12:40:00 PM |
|---|---|---|

| Page 17: [26] Deleted | Gijs de Boer | 10/27/20 12:40:00 PM |
|---|---|---|

| Page 17: [27] Formatted | Gijs de Boer | 10/27/20 12:41:00 PM |
|---|---|---|

Space After: 0 pt, Line spacing: 1.5 lines

| Page 17: [28] Deleted | Gijs de Boer | 10/27/20 12:41:00 PM |
|---|---|---|

| Page 17: [28] Deleted | Gijs de Boer | 10/27/20 12:41:00 PM |
|---|---|---|

| Page 19: [29] Deleted | Gijs de Boer | 11/2/20 8:56:00 AM |
|---|---|---|